# InnoAds: A Foundation Dataset and Benchmark for Advertising Visual Effects in Video Generation

## Abstract

Advertising visual effects (Ad-VFX) are the significant visual elements of advertising videos that combine dynamic product presentation with accompanying descriptive text. However, research on Ad-VFX has been hindered by the lack of dedicated datasets and standardized evaluation protocols, as it is still an emerging domain. To address this issue, we introduce **InnoAds-32K**, a foundation dataset of over 32,000 curated image–text–video triples tailored for advertising scenarios. Furthermore, we propose **InnoAds-Bench**, a comprehensive benchmark that spans six evaluation dimensions: visual quality and text relevance as general metrics, and motion, product consistency, text stability, and creative rationality as advertising-specific metrics. Based on this suite, we systematically evaluate state-of-the-art video generation models, revealing substantial limitations in their ability to satisfy advertising requirements. In summary, InnoAds-32K and InnoAds-Bench provide the first standardized foundation for Ad-VFX video generation, paving the way for future research in advertising scenarios. An anonymous website is available at https://innoads-anon.github.io/InnoAds.github.io/.

## 1 Introduction

Video generation has made remarkable progress in recent years Kwai (2025); Wan et al. (2025), with state-of-the-art (SOTA) models capable of producing high-fidelity content across diverse application scenarios Wang et al. (2025b); Che et al. (2024). However, these advances remain insufficient for advertising scenarios, particularly for **Advertising Visual Effects (Ad-VFX)**, the significant visual elements of advertising videos that combine dynamic product presentation with accompanying descriptive text.

Unlike general scenarios video generation, which is often evaluated primarily on visual quality, text relevance, and motion amplitude and smoothness Yuan et al. (2025), Ad-VFX imposes additional requirements that directly reflect the priorities of advertising practice. Beyond producing visually appealing frames, generated videos must ensure motion continuity, providing sustained and coherent movements throughout the entire video to capture viewer's attention; maintain product consistency to faithfully preserve the identity, shape, and details of the item; guarantee text stability such that brand names, promotional labels, and descriptions remain legible throughout the video; and achieve creative rationality, balancing imaginative visual effects with commercial plausibility. Existing video generation models MiniMax (2025); Kong et al. (2024) struggle to satisfy these requirements, often producing videos with incoherent motion, distorted products, and unstable text, while failing to align with advertising objectives.

A critical bottleneck underlying these challenges is the absence of **advertising-specific datasets** and **evaluation benchmarks**. As shown in Table 1, widely used datasets such as Panda-70M Chen et al. (2024), ActivityNet Caba Heilbron et al. (2015) and OpenHumanVid Li et al. (2024) mainly cover general scenarios including daily activities, natural scenes and human actions, with large scales and diverse resolutions, but none are curated around advertising scenarios. They also lack annotations that are crucial for Ad-VFX, for example detailed product categories, requirements for text clarity and constraints on motion dynamics. These limitations make them inadequate for extending foundation models to Ad-VFX video generation. Similarly, existing evaluation benchmarks summarized in

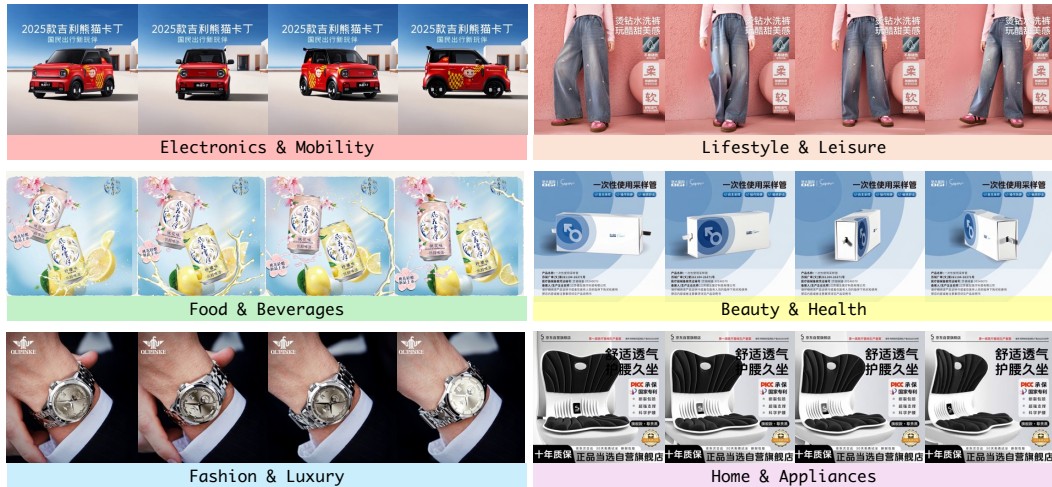

Figure 1: Example cases from InnoAds-32K, illustrating six core product categories.

Table 2 focus on general metrics such as visual quality, text relevance and motion amplitude and smoothness. Although effective in measuring broad generation quality, they consistently overlook dimensions that are indispensable for advertising practice, including motion continuity, product consistency, text stability and creative rationality. This gap prevents systematic evaluation and improvement of video generation models in advertising scenarios.

To address these issues, we present **InnoAds-32K** and **InnoAds-Bench**, the first dataset–benchmark suite dedicated to Ad-VFX. InnoAds-32K is a collection of over 32,000 curated image–text–video triples focused on advertising scenarios. It emphasizes product-centric content, with clear textual annotations and diverse motion patterns, and further covers a broad spectrum of advertising domains. Specifically, the dataset spans 44 fine-grained categories organized into six major domains: *Electronics & Mobility*, *Lifestyle & Leisure*, *Food & Beverages*, *Beauty & Health*, *Fashion & Luxury*, and *Home & Appliances*, as shown in Figure 1. This diversity provides the necessary resources for training and adapting video generation models to multifaceted requirements of advertising practice.

Furthermore, we introduce InnoAds-Bench, a comprehensive benchmark designed to evaluate Ad-VFX along six complementary dimensions. Specifically, AesScore Schuhmann (2024) and GmeScore Yuan et al. (2025) serve as general metrics for visual quality and text relevance, while MotionXScore, ProductScore, TSSScore, and CRScore target advertising-specific requirements by assessing motion, product consistency, text stability, and creative rationality. In addition, InnoAds-Bench consists of 176 curated images from 44 categories, with four representative images per category, ensuring diverse coverage of advertising scenarios. In summary, InnoAds-32K and InnoAds-Bench together establish a standardized foundation for systematic research on Ad-VFX, bridging the gap between general scenarios video generation studies and those focused on advertising scenarios.

Based on this suite, we systematically evaluate SOTA video generation models, covering four closed-source models, four open-source models, and two fine-tuned models by InnoAds-32K. Our evaluation reveals that existing models remain inadequate for advertising scenarios. Moreover, experiments demonstrate that fine-tuning on InnoAds-32K significantly enhances model performance under advertising-specific requirements, underscoring the practical value of our dataset and benchmark.

The contributions of this work are as follows:

**i) InnoAds-32K.** We present InnoAds-32K, the foundation dataset of over 32,000 curated image–text–video triples tailored to Ad-VFX, covering six major domains.

**ii) InnoAds-Bench.** We introduce InnoAds-Bench, a comprehensive benchmark that evaluates Ad-VFX along six dimensions, including both general metrics and advertising-specific metrics.

**iii) Experiments.** We conduct extensive experiments on InnoAds-Bench, demonstrating that existing approaches fall short in advertising scenarios while our suite enable notable improvements.

Table 1: Comparison of the Statistics of InnoAds-32K with existing Video Generation Datasets. Most of them are inadequate for extending foundational models to video generation task.

| Dataset | Scene | Resolution | Video Clips | Average Length (s) | Video Duration (h) |
|---|---|---|---|---|---|
| Panda-70M Chen et al. (2024) | Open | 720P | 70M | 8.6 | 167K |
| ChronoMagic-Pro Yuan et al. (2024) | Time-Lapse | 720p | 460K | 234.8 | 30K |
| OpenHumanVid Li et al. (2024) | Human | 720P | 52.3M | 4.9 | 70K |
| ActivityNet Caba Heilbron et al. (2015) | Action | - | 85K | 118.5 | 849 |
| YouCook2 Zhou et al. (2018) | Cooking | - | 32K | 19.6 | 176 |
| How2 Sanabria et al. (2018) | Instruct | - | 191K | 90 | 308 |
| OGameData Che et al. (2024) | Game | 720P-4K | 1000K | - | 4K |
| OpenS2V-5M Yuan et al. (2025) | Subject | 720P | 5.4M | 6.6 | 10K |
| **InnoAds-32k** | Product | $960 \times 960$ | 32K | 5.0 | 44.4 |

Table 2: Comparison of the Characteristics of our InnoAds-Bench with existing Benchmarks. Most of them focus on T2V and neglect the evaluation of Ad-VFX. _ means suboptimal for Ad-VFX.

| Benchmark | Visual Quality | Text Relevance | Motion | Product Consistency | Text Stability | Creative Rationality |
|---|---|---|---|---|---|---|
| Make-a-Video-Eval Singer et al. (2022) | ✓ | ✓ | ✗ | ✗ | ✗ | ✗ |
| FETV Liu et al. (2024b) | ✓ | ✓ | ✓ | ✗ | ✗ | ✗ |
| T2VScore Wu et al. (2024) | ✓ | ✓ | ✓ | ✗ | ✗ | ✗ |
| EvalCrafter Liu et al. (2024a) | ✓ | ✓ | ✓ | ✗ | ✗ | ✗ |
| VBench Huang et al. (2023) | ✓ | ✓ | ✓ | ✗ | ✗ | ✗ |
| VBench++ Huang et al. (2024) | ✓ | ✓ | ✓ | ✗ | ✗ | ✗ |
| A2 Bench Fei et al. (2025) | ✓ | ✓ | ✓ | ✓ | ✗ | ✗ |
| VACE-Bench Jiang et al. (2025) | ✓ | ✓ | ✓ | ✓ | ✗ | ✗ |
| OpenS2V-Eval Yuan et al. (2025) | ✓ | ✓ | ✓ | ✓ | ✗ | ✗ |
| **InnoAds-Bench** | ✓ | ✓ | ✓ | ✓ | ✓ | ✓ |

## 2 RELATED WORK

**Datasets for Visual Generation.** Large-scale image generation datasets such as Laion-5B Schuhmann et al. (2022) and DiffusionDB Wang et al. (2022) have driven progress in general domains, while advertising-specific resources are still emerging. Representative efforts include BG60k Wang et al. (2025a) for e-commerce product background generation, PPG30k for poster generation with layout and text annotations, and RF1M Du et al. (2024) with over one million product images. These datasets advance image-level tasks but remain limited to static content. In contrast, Ad-VFX generation suffers from the lack of dedicated video datasets. Existing large-scale video corpora, such as Panda-70M Chen et al. (2024), OpenVid-1M Nan et al. (2024), and WebVid-10M Xiong et al. (2024), cover general scenarios but fail to capture advertising-specific requirements. This scarcity of specialized video datasets has become a major bottleneck for developing effective Ad-VFX models.

**Automatic Metrics for Video Generation.** With the rapid advancement of generative models, several benchmarks have been developed to evaluate the quality of video generation. VBench Huang et al. (2023; 2024) dissects video quality into structured dimensions with tailored prompts, EvalCrafter Liu et al. (2024a) provides a large prompt set reflecting real-world user data, and VACE-Bench Jiang et al. (2025) curates diverse videos for evaluating multiple generation and editing tasks. Although these benchmarks cover important aspects such as visual fidelity, temporal consistency, and text relevance, they are designed for general scenarios and overlook requirements that are critical in advertising, such as motion continuity, product consistency, textual stability and creative rationality.

## 3 INNOADS-32K

### 3.1 DATASET CONSTRUCTION

To construct InnoAds-32K, we develop a multi-stage pipeline to ensure the quality and reliability of advertising-specific video content, as illustrated in Figure 2.

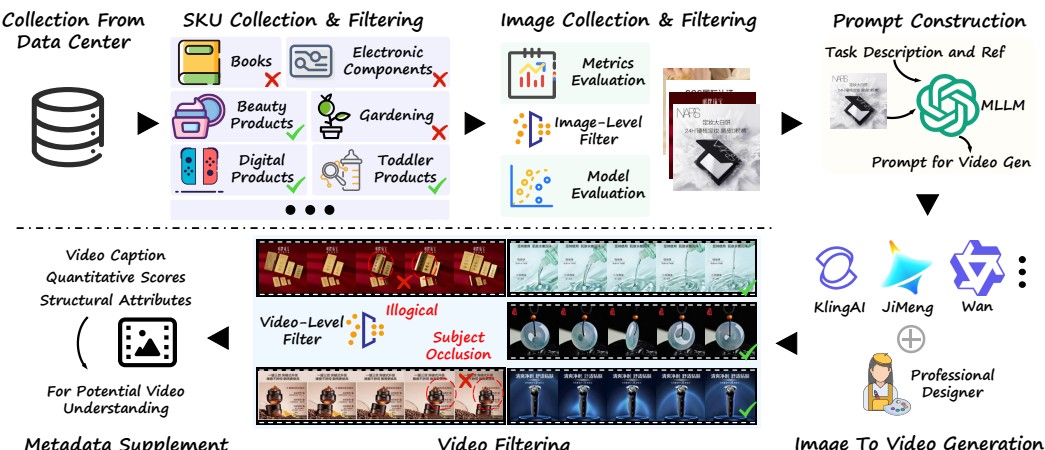

Figure 2: The multi-stage pipeline of InnoAds-32K, consisting of SKU collection and filtering, image collection and filtering, prompt construction, image-to-video generation, video filtering, and metadata supplementation.

**SKU Collection and Filtering.** We begin by collecting stock-keeping unit (SKU) categories from major e-commerce platforms. To prioritize products that are motion-rich and visually expressive, categories with inherently static or intangible characteristics are excluded. For instance, household items with minimal visual variation and digital goods without a tangible form are removed. This filtering step ensures that the retained categories remain relevant to advertising contexts and are well-suited for dynamic video generation.

**Image Collection and Filtering.** For each retained SKU, representative advertising images are collected to support prompt construction and subsequent video generation. As the raw collection inevitably contains low-quality samples and content misaligned with advertising needs, we apply a multi-perspective filtering pipeline to ensure data quality and scenario relevance. The filtering process evaluates images along four complementary dimensions: image quality, layout structure, subject saliency, and text interference. Accordingly, we remove visually degraded samples, exclude non-standard or collage-style layouts, enforce clear visual focus on the target SKU, and filter out cases where excessive or occluding text compromises product visibility. Beyond these heuristic rules, we further combine the filtered subset with a portion of manually annotated samples to train a binary classification model using CLIP Radford et al. (2021) and DINOv3 Siméoni et al. (2025) features, enabling an additional layer of automated quality control. This process yields a curated collection of high-quality, content-rich, and product-centric images that form a reliable foundation for the subsequent stages of dataset construction.

**Prompt Construction.** We employ the multi-modal large language model GPT-5 OpenAI (2025) to construct prompts via structured queries that combine images with tailored instructions. The generated prompts emphasize plausible product motions and dynamic cues, ensuring that the textual guidance facilitates the synthesis of temporally coherent and advertising-relevant video sequences. Details of the structured queries are provided in Appendix C.5.

**Image-to-Video Generation.** The motion-aware prompts and corresponding SKU images are processed by SOTA image-to-video models, including Kling Kwai (2025), Jimeng ByteDance (2025), and Wan Wan et al. (2025), to generate candidate advertising videos. These models render products with realistic and visually coherent motion dynamics. To further ensure quality and maintain diversity, a subset of videos is curated and refined with input from professional designers, thereby complementing automated synthesis with human expertise. This hybrid strategy leverages the scalability of generative models together with the precision of manual curation, resulting in a reliable and representative video collection.

**Video Filtering.** The raw generated videos undergo a dedicated filtering stage to ensure data quality. A multi-perspective evaluation framework is employed, incorporating quantitative metrics introduced in Section 4. This framework enables the detection of common issues, including insufficient or

unstable motion, degraded or illegible text, reduced visual quality, and inconsistencies between the SKU image and its video representation. Only videos that satisfy these criteria are retained, ensuring that the dataset remains both reliable and suitable for downstream research tasks.

**Metadata Supplement.** In the final stage, each advertising video is enriched with auxiliary metadata to enhance its research utility. The metadata include captions generated by Qwen2.5-VL Bai et al. (2025) that describe scene content and product dynamics, quantitative scores assessing dimensions such as motion and aesthetics, and structural attributes such as resolution and frame count. These additional metadata improve the interpretability of the dataset and facilitate broader research directions, thereby supporting tasks in multi-modal learning and video understanding.

## 3.2 DATASET STATISTICS

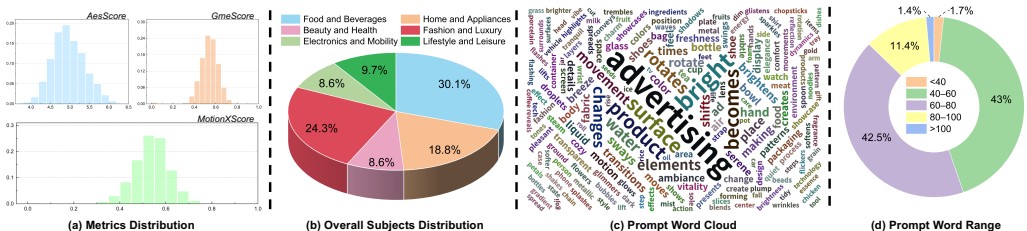

(a) Metrics Distribution  (b) Overall Subjects Distribution  (c) Prompt Word Cloud  (d) Prompt Word Range

Figure 3: Statistics of InnoAds-32K, including (a) video quality evaluation via InnoAds-Bench, (b) balanced distribution across six major e-commerce categories, (c) word cloud of advertising-related prompts, and (d) prompt length distribution.

InnoAds-32K is the first high-quality dataset specifically designed for advertising scenarios. It is built through a systematic multi-stage pipeline that incorporates domain-specific generation and filtering strategies to align with advertising requirements. Figure 3 reports key statistics of InnoAds-32K, including video-level quality measures and characteristics of the associated textual prompts.

For video quality, we adopt InnoAds-Bench metrics (Figure 3(a)) to filter out low-quality samples, ensuring reliable training data. The dataset maintains a balanced distribution across six e-commerce domains, *Food & Beverages* (30.1%), *Fashion & Luxury* (24.3%), *Home & Appliances* (18.8%), *Lifestyle & Leisure* (9.7%), *Electronics & Mobility* (8.6%), and *Beauty & Health* (8.6%), with 44 subcategories detailed in Appendix A. Beyond categorical balance, InnoAds-32K provides rich textual prompts (Figures 3(c,d)), where frequent tokens capture both product attributes and advertising cues, and most prompts fall between 40–80 tokens.

## 4 INNOADS-BENCH

### 4.1 BENCHMARK STATISTICS

InnoAds-Bench comprises 176 cases across six major domains. The distribution includes *Home & Appliances* with 44 cases (25%), *Food & Beverages* with 32 (18.8%), both *Beauty & Health* and *Fashion & Luxury* with 28 each (15.91%), *Electronics & Mobility* with 24 (13.64%), and *Lifestyle & Leisure* with 20 (11.36%), as shown in Figure 4 The distribution is derived from real user purchasing statistics over a representative time period, reflecting practical advertising priorities in which household goods and consumables dominate, while lifestyle products remain consistently represented. Further analyses of InnoAds-Bench are provided in Appendix B.1.

### 4.2 METRICS FOR AD-VFX

Existing video generation metrics mainly target general scenarios, emphasizing visual quality and text relevance, but they are insufficient for Ad-VFX. Our framework addresses this gap by spanning six dimensions: two general ones evaluated with AesScore and GmeScore, and four advertising-specific ones measured by MotionXScore, ProductScore, TSScore, and CRScore. These metrics provide a comprehensive basis for assessing advertising scenarios video generation.

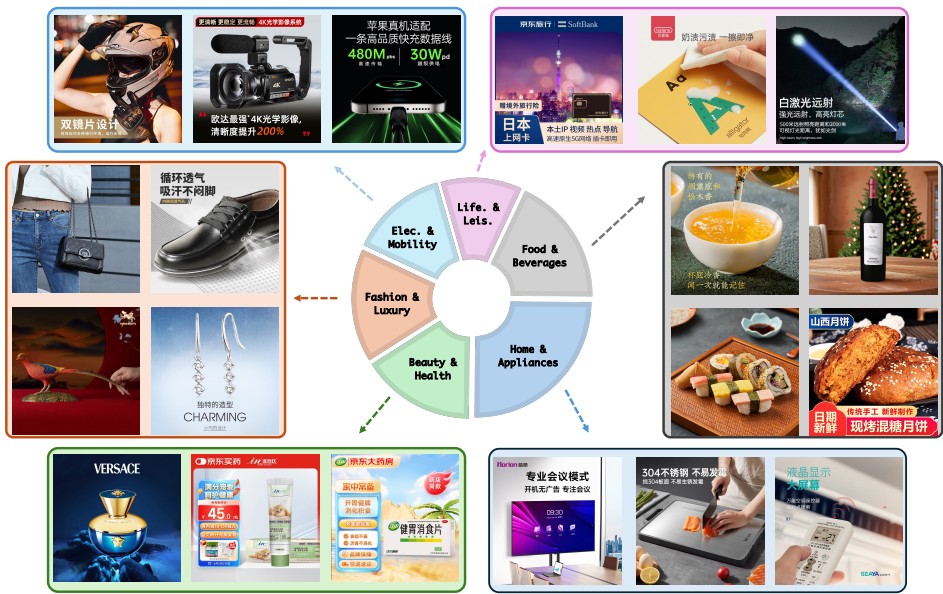

Figure 4: Representative advertising samples from six major domains in InnoAds-Bench.

**MotionXScore.** Motion continuity is particularly crucial for Ad-VFX, as static or weakly animated content often fails to capture attention or convey product dynamics. Prior work such as OpenS2V Yuan et al. (2025) defines MotionScore based on *amplitude* and *smoothness*, but these measures can be overly optimistic when only a few frames contain significant movement. To address this limitation, we extend the metric with an additional *activation ratio*, which explicitly measures the persistence of motion throughout the video, thereby providing a more faithful evaluation for advertising scenarios.

Specifically, given per-frame motion strength $E_t$ Lucas & Kanade (1981), we first apply temporal smoothing and then determine an adaptive threshold using the median absolute deviation (MAD):

$$\tau = \text{Median}(\tilde{E}_t) + k \cdot (1.4826\,\text{MAD}), \quad \text{MAD} = \text{Median}\left(\left|\tilde{E}_t - \text{Median}(\tilde{E}_t)\right|\right). \quad (1)$$

We adopt an adaptive threshold rather than a fixed one, since motion magnitude varies widely across videos. A fixed threshold either overlooks subtle yet meaningful movements in static videos or classifies nearly all frames as active in highly dynamic ones. By normalizing against each video's MAD, our method distinguishes motion that rises significantly above background fluctuations, yielding a fair and robust measure of motion persistence. A frame is marked as "active" if $\tilde{E}_t \geq \tau$, and we enforce a minimum run length $L_{\min}$ to filter out isolated spikes. The *activation ratio* is then defined as

$$R = \frac{1}{T-1} \sum_{t=1}^{T-1} \hat{z}_t, \quad \hat{z}_t = \mathbb{1}\left[\tilde{E}_t \geq \tau \,\wedge\, |\text{run}(t)| \geq L_{\min}\right], \quad (2)$$

where $\mathbb{1}\,[\cdot]$ is the indicator function. Finally, MotionXScore normalizes the three components, motion amplitude A, smoothness S, and activation ratio R, and computes their average:

$$S_{\text{MotionX}} = \frac{1}{3}\left(\hat{A} + \hat{S} + \hat{R}\right), \quad (3)$$

where $\hat{A}, \hat{S}, \hat{R}$ denote normalized values of amplitude, smoothness, and activation ratio, respectively.

**ProductScore.** The ProductScore is built on NexusScore Yuan et al. (2025), which evaluates subject–video alignment by detecting the reference object in generated frames and computing similarity with a multi-modal embedding model Zhang et al. (2024). The aggregated similarity reflects how well the reference input is preserved in the video. However, advertising images often include substantial textual and decorative content, which introduces noise when assessing product consistency. To address this, we employ Grounded-SAM2 Ravi et al. (2024); Liu et al. (2023) guided

by the product name to detect and segment the advertised item from the reference image. The resulting subject crop serves as the visual prompt in NexusScore, ensuring that evaluation focuses exclusively on the product itself. ProductScore therefore extends NexusScore by explicitly incorporating product crops, yielding a more faithful measure of identity and appearance preservation in Ad-VFX scenarios while reducing confounding from textual or background artifacts.

**TSScore.** Unlike metrics that evaluate text–prompt alignment, advertising videos require assessment of visual text within the video. Such text is central to the message but often disappears, deforms, or becomes unreadable. TSScore addresses this by measuring stability and legibility across three aspects, providing a dedicated metric for advertising scenarios.

To avoid interference from textual elements printed on the product (e.g., logos or packaging), we first exclude the detected product region based on ProductScore preprocessing, and only retain the surrounding areas for text evaluation. Given a set of sampled frames $\{F_t\}_{t=1}^T$, we apply OCR Cui et al. (2025) to detect text bounding boxes and recognized strings. Detected boxes are grouped across frames into text segments $\mathcal{L} = \{L_j\}$ by combining spatial overlap and textual similarity. For each segment $L_j$, we first measure content stability through the cross-frame consistency of recognized strings. Let $T_t$ denote the text in frame $t$, the stability is defined as

$$\text{Stab}(L_j) = 1 - \frac{1}{|L_j| - 1} \sum_t \text{CER}(T_t, T_{t+1}), \tag{4}$$

where CER is the character error rate and higher values indicate more consistent recognition. We then measure clarity to capture the visual sharpness and legibility of text. For each bounding box $b_t$, we combine OCR confidence $c_t$ with an image-based sharpness measure, namely the variance of Laplacian, as

$$\text{Clarity}(b_t) = \lambda \cdot c_t + (1 - \lambda) \cdot \text{NormVarLap}(b_t), \tag{5}$$

and average over all frames in the segment. Finally, we quantify temporal coverage to reflect how persistently a text remains visible in the video:

$$\text{Cover}(L_j) = \frac{|L_j|}{T}, \tag{6}$$

where $|L_j|$ is the number of frames in which the text is detected. The segment-level scores are integrated into the overall metric by weighted averaging:

$$S_{\text{Text}} = \frac{1}{|\mathcal{L}|} \sum_{j \in \mathcal{L}} \left( \alpha \cdot \text{Stab}(L_j) + \beta \cdot \text{Clarity}(L_j) + \gamma \cdot \text{Cover}(L_j) \right), \tag{7}$$

where $\alpha, \beta, \gamma$ are trade-off parameters. By explicitly removing product regions and focusing on surrounding texts, TSScore provides a dedicated measure of the stability and readability of advertising-related texts in generated videos.

**CRScore.** In addition to fidelity and stability, advertising videos must balance creativity with rationality: the generated content should be sufficiently imaginative to capture attention while remaining coherent and plausible to viewers. These aspects are difficult to capture with traditional low-level metrics. We therefore introduce CRScore, a metric that leverages a powerful MLLM OpenAI (2025) to automatically assess the creative rationality of generated videos.

Given a video $\{F_t\}_{t=1}^T$, we uniformly sample frames to form a concise visual summary, which is fed into the MLLM together with a natural-language prompt instructing the model to judge whether the video demonstrates creative visual effects while preserving logical plausibility. The model outputs a discrete score $s^{(k)} \in [1, 5]$ for the $k$-th evaluation. To reduce stochasticity, the evaluation is repeated $N$ times and the final score is computed as the mean:

$$S_{\text{CR}} = \frac{1}{N} \sum_{k=1}^N s^{(k)}. \tag{8}$$

This design extends evaluation beyond pixel-level fidelity, capturing high-level dimensions of creativity and rationality that are critical in Ad-VFX scenarios. Details of the prompt design and implementation are provided in the Appendix C.5.

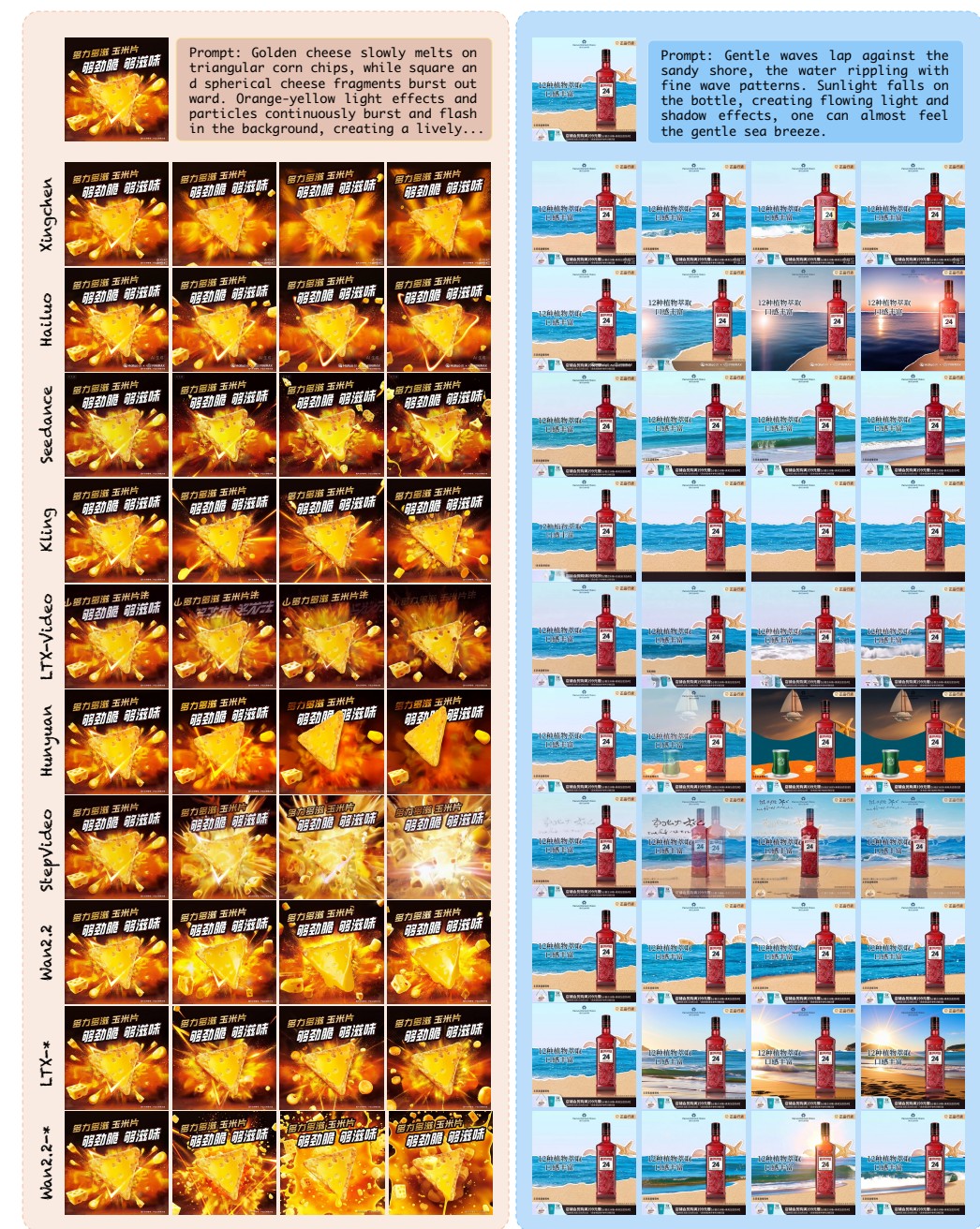

Figure 5: Qualitative comparison of representative video frames generated under InnoAds-Bench prompts. We compare closed-source models, open-source models, and our fine-tuned baselines.

# 5 EXPERIMENTS

## 5.1 EVALUATION SETUPS

**Baselines.** We evaluate a diverse set of SOTA video generation models on InnoAds-Bench, encompassing both closed-source and open-source models. Specifically, the baselines include Taobao Xingchen Taobao (2025), Hailuo 02 MiniMax (2025), Seedance 1.0 mini ByteDance (2025), Kling 2.1 Std Kwai (2025), LTX-Video-13B HaCohen et al. (2024), HunyuanVideo Kong et al. (2024), StepVideo Ma et al. (2025), and Wan2.2-A14B Wan et al. (2025). To further validate the effectiveness

Table 3: Quantitative comparison on InnoAds-Bench. The best scores are **bolded**, while the second-best is underlined.

| Type | Method | AesScore↑ | GmeScore↑ | MotionX↑ | ProductScore↑ | TSScore↑ | CRScore↑ |
|------|--------|-----------|-----------|----------|---------------|----------|----------|
| Closed | Taobao Xingchen Taobao (2025) | 26.90% | **59.03%** | 54.33% | 47.31% | 78.26% | 60.31% |
| | Hailuo 02 MiniMax (2025) | 27.11% | 58.32% | 55.13% | 48.42% | 77.36% | 65.14% |
| | Seedance 1.0 mini ByteDance (2025) | 27.52% | 57.17% | 53.02% | 45.16% | 75.19% | 62.00% |
| | Kling 2.1 Std Kwai (2025) | **31.76%** | 58.14% | **56.58%** | 50.64% | **80.80%** | 66.25% |
| Open | LTX-Video-13B HaCohen et al. (2024) | 20.20% | 56.83% | 49.11% | 33.79% | 64.70% | 54.87% |
| | HunyuanVideo Kong et al. (2024) | 28.16% | 54.50% | 48.01% | 39.31% | 69.76% | 54.19% |
| | StepVideo Ma et al. (2025) | 20.68% | 53.22% | 48.31% | 31.05% | 63.84% | 52.50% |
| | Wan2.2-A14B Wan et al. (2025) | 29.05% | 56.78% | 48.34% | 41.86% | 72.82% | 60.65% |
| Finetuned | LTX-InnoAds | 27.93% | 58.02% | 50.01% | 40.06% | 70.95% | 57.10% |
| | Wan2.2-InnoAds | 31.47% | 58.90% | 54.05% | **50.71%** | 78.42% | **67.23%** |

of our dataset, we additionally fine-tune two representative models, LTX and Wan2.2, on InnoAds-32K, resulting in LTX-InnoAds and Wan2.2-InnoAds. More details of these models are provided in the Appendix C.1.

**Implementation Details.** For closed-source models, we query the official interfaces with the same prompts and input images, ensuring consistent evaluation across models. For open-source models including Wan2.2, LTX, HunyuanI2V, and StepVideo, we employ the officially released weights and hyper-parameters, and run all experiments on an 8×NVIDIA H800 GPU cluster. All results are reported as averages over 32 test samples, with multiple random seeds to mitigate variance. Further details are provided in the Appendix C.2.

## 5.2 COMPREHENSIVE ANALYSIS

**Quantitative Evaluation.** Table 3 reports results on InnoAds-Bench across six metrics. Among closed-source models, Kling performs best overall, leading in *AesScore* (31.76%), *MotionX* (56.58%), and *TSScore* (80.80%), while also ranking second in *ProductScore* and *CRScore*. Taobao Xingchen achieves the highest *GmeScore* (59.03%), and Hailuo remains competitive in *MotionX* and *CRScore*. Open-source models lag behind: HunyuanVideo shows a relatively higher *AesScore* (28.16%), but most struggle with product consistency and text stability. Fine-tuning on InnoAds-32K yields consistent gains, Wan2.2-InnoAds achieves the best *ProductScore* (50.71%) and *CRScore* (67.23%) and ranks second on several other metrics, while LTX-InnoAds surpasses its base model across all dimensions. These results highlight both the limitations of current models and the clear benefits of domain-specific adaptation. More details are provided in the Appendix C.4.

**Qualitative Evaluation.** Figure 5 presents qualitative comparisons on two representative cases: one focused on advertising effects and the other on product presentation. In the first case, closed-source models generally preserve text but show artifacts, such as wave lines in Xingchen and melting textures in Seedance. Open-source models perform worse, with LTX losing visual text, StepVideo producing excessive motion, and Wan2.2 altering product identity. By contrast, the fine-tuned LTX-InnoAds and Wan2.2-InnoAds retain text and generate coherent motion. In the second case, most closed-source models yield weak or static motion, with Hailuo providing relatively natural dynamics; open-source models again suffer from limited motion or distortions. Fine-tuned models consistently maintain textual stability while delivering reasonable dynamics, underscoring both the shortcomings of current methods and the benefits of adaptation with InnoAds-32K.

## 6 CONCLUSION

In this paper, we presents InnoAds-32K, the foundation dataset specifically curated for Ad-VFX, and InnoAds-Bench, a comprehensive benchmark that evaluates Ad-VFX across six dimensions, including both general and advertising-specific metrics. Through extensive experiments, we reveal that existing SOTA models still struggle with advertising-specific requirements. Moreover, we demonstrate that fine-tuning on InnoAds-32K yields consistent improvements across multiple metrics, highlighting the dataset's value for domain adaptation. We hope that InnoAds-32K and InnoAds-Bench will serve as standardized resources to drive further advances in advertising-oriented video generation, bridging the gap between general video synthesis and real-world advertising applications.

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

# APPENDIX: SUPPLEMENTARY MATERIAL

## A   MORE DETAILS OF INNOADS-32K

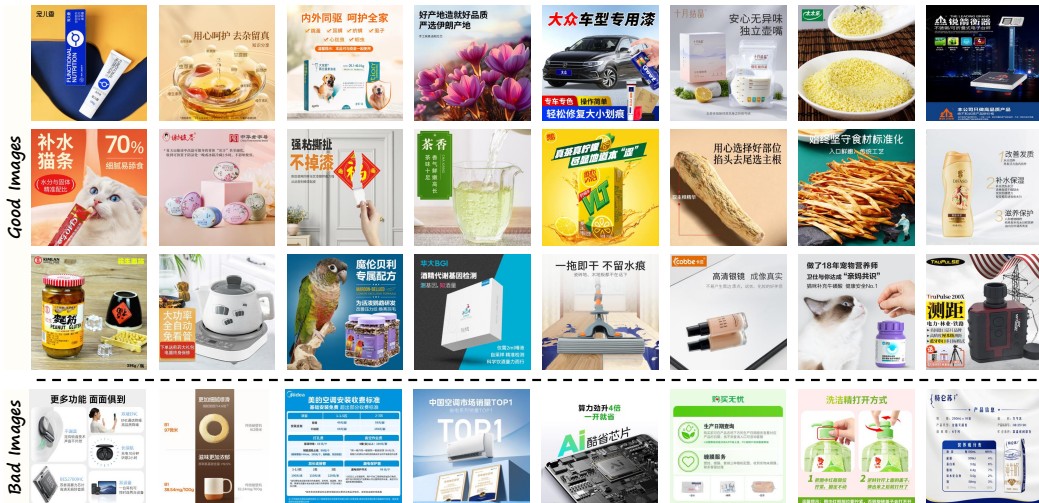

Figure 6: Examples of image quality control in the InnoAds-32K dataset. The upper row shows *Good Images* retained after filtering, while the lower row presents *Bad Images* that were discarded.

Figure 6 presents representative image samples used in the construction of InnoAds-32K. The dataset incorporates a rigorous filtering pipeline to ensure the high quality of advertising content. As illustrated, the upper part shows examples of images retained in the dataset (*Good Images*), characterized by clear product presentation, well-designed layouts, and sharp, readable text. In contrast, the lower part shows examples of filtered-out cases (*Bad Images*), which often suffer from poor visual design, excessive textual clutter, or low informativeness. This quality control step guarantees that the dataset emphasizes visually appealing, advertisement-ready content while systematically excluding noisy or suboptimal inputs.

Figure 7 presents the fine-grained category distribution of InnoAds-32K. In addition to the six major e-commerce domains (*Food and Beverages*, *Home and Appliances*, *Beauty and Health*, *Fashion and Luxury*, *Electronics and Mobility*, and *Lifestyle and Leisure*), the dataset is further divided into 44 subcategories. These subcategories capture the diversity of real-world advertising contexts, covering both essential consumer goods (e.g., packaged food, personal care, mobile phones) and high-value product lines (e.g., jewelry, luxury goods, artworks). Such hierarchical organization ensures that the dataset not only achieves balanced coverage at the domain level but also provides sufficient granularity for downstream tasks, enabling models to better adapt to heterogeneous product characteristics and category-specific advertising requirements.

## B   MORE DETAILS OF INNOADS-BENCH

### B.1   DESIGN AND ANALYSIS OF INNOADS-BENCH

Diverse test cases are essential for InnoAds-Bench to function as a rigorous evaluation tool for product video generation. Unlike general benchmarks, product-focused assessment must address category-specific visual requirements: *Food & Beverages* emphasize texture and freshness, *Home & Appliances* require demonstrations of functionality, and *Fashion & Luxury* demand accurate portrayal of materials and aesthetics. Without such coverage, evaluations risk bias, as models that perform well on rigid-surfaced electronics may fail on fabrics or cosmetics. By spanning perishable items to luxury goods, InnoAds-Bench evaluates a model's ability to adapt to varied visual grammars.

This comprehensive benchmark design offers key advantages for reliable model evaluation. First, it eliminates category bias, requiring models to demonstrate proficiency across 176 diverse cases

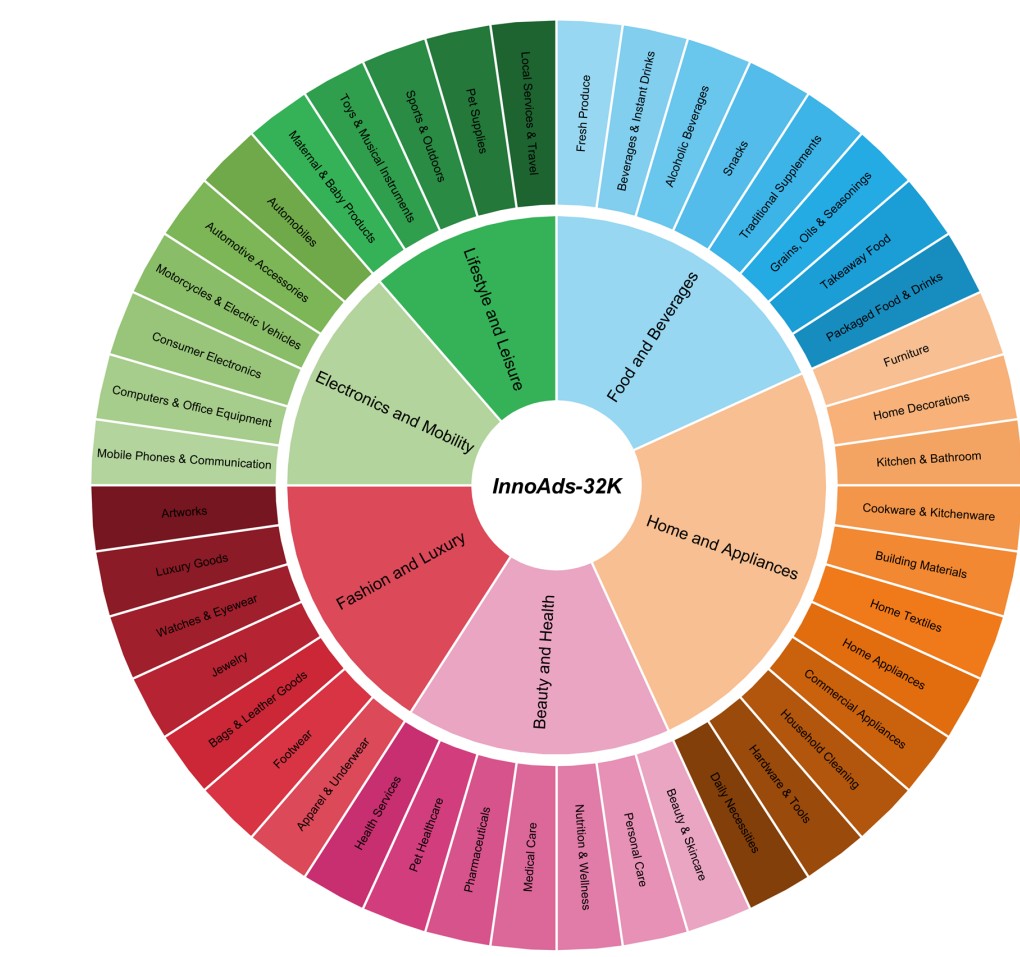

Figure 7: Fine-grained category distribution of the InnoAds-32K dataset, illustrating 44 subcategories grouped under six major domains.

to ensure generalizability to real-world scenarios. Second, it enables granular analysis, revealing whether a model excels at rendering small gadgets but struggles with large furniture, or performs well on cosmetics while failing on food textures. Third, its composition reflects real commercial content, ensuring that evaluation results remain ecologically valid and directly applicable to advertising practice. By avoiding narrow sampling, InnoAds-Bench provides a robust measure of a model's capability across the full spectrum of product video generation tasks.

### B.2 DETAILS OF AESSCORE AND GMESCORE

Both AesScore and GmeScore are adapted from the metric design of OpenS2V Yuan et al. (2025).

**AesScore.** AesScore measures the aesthetic quality of generated frames. Following OpenS2V, it combines CLIP embeddings with an additional MLP head trained for aesthetic prediction. This design enables the metric to capture both semantic alignment and aesthetic preferences.

**GmeScore.** GmeScore evaluates text–video relevance. Existing approaches typically rely on CLIP Radford et al. (2021) or BLIP Zhang et al. (2023), but prior work has shown that these models suffer from feature space inconsistencies and limited text encoders (restricted to 77 tokens), making them unsuitable for the longer prompts favored by modern DiT-based video generation models. To address this, OpenS2V replaces them with GME Zhang et al. (2024), a model fine-tuned on Qwen2-VL Wang et al. (2024). This choice allows GmeScore to handle variable-length prompts and produce more reliable relevance scores.

Figure 8: Multi-dimensional evaluation metrics for AD-VFX in InnoAds-Bench.

### B.3 Illustrations of Metric Design

Figure 8 provides schematic illustrations of our evaluation metrics, highlighting their inputs and outputs for clarity.

## C More Details of Experiment

### C.1 Details of Evaluation Models

For completeness, we provide here additional details of the baseline models evaluated on InnoAds-Bench. These include both closed-source and open-source models, as well as their fine-tuned variants on InnoAds-32K. The descriptions below briefly summarize each model's architecture and application background.

**Taobao Xingchen Taobao (2025).** Taobao Xingchen is a commercial-grade video generation system launched by Alibaba's Alimama team, specifically tailored for e-commerce advertising scenarios. Given a single product image and a short text prompt, it can synthesize high-quality short videos (around 5 seconds) enriched with advertising elements such as lighting effects, transitions, visual effects, and text overlays. Unlike general-purpose video generation models, Xingchen emphasizes product fidelity, semantic alignment, frame stability, and controllable dynamics, aiming to reduce issues such as jitter, distortion, or occlusion. Its technical foundation builds on Alibaba's proprietary large-model architecture (e.g., the Tbstar series) and integrates domain-specific e-commerce data, design language, and marketing knowledge, thereby enabling video generation that is more product-aware and better aligned with commercial advertising needs.

**Kling Kwai (2025).** Kling is a commercial text-to-video system developed by Kuaishou (Kwai), and version 2.1 is one of its latest releases. The system supports both image-to-video and prompt-driven generation, producing short video clips (e.g. 5 s) at resolutions up to 1080p. According to public sources, Kling 2.1 offers dual modes: a "Standard" mode at 720p and a "Professional" mode at 1080p, allowing users to trade off between speed and quality. Kuaishou claims that Kling is capable of generating videos up to two minutes long, at 30 fps, with support for multiple aspect ratios and cinematic control of camera movement. Unlike many purely research-grade models, Kling is intended for commercial use and emphasizes realistic motion, visual fidelity, and prompt adherence under practical constraints.

**Seedance ByteDance (2025).** Seedance is a video foundation model developed by ByteDance that supports both text-to-video and image-to-video generation. It can produce short cinematic clips (e.g. 5 s at 1080p) with smooth motion, rich structure, and prompt adherence. Its architecture features multi-shot narrative coherence, interleaved multi-modal positional encoding, and optimization strategies such as RLHF and multi-stage distillation that accelerate inference speed by 10×. Compared to other models, Seedance emphasizes both motion fluidity and structural stability under complex prompts.

**Hailuo MiniMax (2025).** Hailuo is a closed-source video generation system developed by MiniMax. It is tailored for creative advertising and aims to produce dynamic and aesthetically appealing content with strong motion rendering. Its marketing emphasizes artistic visual effects and practical applicability in short-form video ads.

**LTX-Video HaCohen et al. (2024).** LTX-Video is an open-source hybrid model that combines diffusion and transformer-based components, designed for high-resolution text-to-video generation. It seeks to balance visual fidelity, temporal coherence, and prompt alignment by leveraging diffusion sampling guided by transformer attention over frames.

**HunyuanVideo (HunyuanVideo-I2V) Kong et al. (2024).** HunyuanVideo-I2V is Tencent's open-source image-to-video extension of the Hunyuan foundation model family. The framework is built on PyTorch and provides LoRA-based fine-tuning capabilities for special effects. It applies token replacement techniques to inject reference image information into the generation process, ensuring consistency with the first frame.

**StepVideo Ma et al. (2025).** StepVideo is a research-oriented open-source model emphasizing step-wise refinement in video generation. Instead of generating full frames in one go, it progressively refines motion and image quality across discrete steps, offering more controllability over motion trajectories and artifacts.

**Wan Wan et al. (2025).** Wan2.2 is an open video generation model open-sourced by Alibaba Tongyi, building on the Wan series. It introduces a Mixture-of-Experts (MoE) architecture, combining high-noise and low-noise expert paths to generate higher-quality video. Wan2.2 also features enhanced training data, aesthetic-aware optimization, and improvements in compression and inference for large-scale deployment.

## C.2 Additional Details of Implementations

For closed-source models, we generate videos via their official APIs or web interfaces. Taobao Xingchen restricts prompt length to 100 tokens; when prompts exceed this limit, we apply truncation with manual refinement to preserve semantic integrity. All closed-source models are evaluated under their default output settings (5s duration, 720p or 960×960 resolution, 81 frames).

For open-source models, we adopt the official checkpoints and inference parameters released by their authors. Experiments are run on 8×NVIDIA A100 GPUs. We follow the standard generation settings of each model to ensure fairness and reproducibility.

For fine-tuned models, we employ LoRA adaptation on both LTX-Video and Wan2.2 using InnoAds-32K. LoRA modules are inserted into the transformer attention layers with rank $r = 64$, and only video backbone parameters are updated while text encoders remain frozen. Training is performed with AdamW (learning rate $1 \times 10^{-4}$, batch size 16). Videos are generated with the same resolution (960×960) and frame settings as the baselines to ensure comparability.

## C.3 Additional Details of Metrics

Several evaluation metrics in InnoAds-Bench involve hyper-parameters that must be specified. For reproducibility, we summarize the settings for MotionXScore, TSScore, and CRScore below.

**MotionXScore.** MotionXScore evaluates motion persistence using activation ratio. We smooth motion energy $E_t$ with an EMA factor $\alpha = 0.5$, normalize per-frame values by the 95th percentile ($p_{\text{inner}} = 95$), and set the activation threshold based on the left-tail quantile ($p_{\text{left}} = 0.3$) plus a scaled MAD factor $\lambda = 0.5$. The minimum active run length is constrained to $L_{\text{min}} = 0.3$ seconds.

Table 4: Hyperparameter settings for MotionXScore.

| Parameter | Value | Description |
|---|---|---|
| $\alpha$ | 0.5 | EMA smoothing factor |
| $p_{\text{inner}}$ | 95 | Per-frame percentile for $E_t$ |
| $p_{\text{left}}$ | 0.3 | Left-tail quantile for threshold |
| $\lambda$ | 0.5 | MAD scaling factor for threshold |
| $L_{\min}$ | 0.3s | Minimum active run length |

**TSScore.** TSScore evaluates the stability of visual text. We set the minimum OCR confidence to 0.5 (ref_conf_thresh), require a similarity score $\geq 0.5$ (keep_sim_thresh) to retain detections, and link bounding boxes across frames if IoU $\geq 0.3$ (iou_link_thresh). Each video is uniformly sampled to 32 frames. Final stability is computed as a weighted sum of three components with $\alpha = \beta = \gamma = \frac{1}{3}$, and text clarity is regularized by $\lambda = 0.7$.

Table 5: Hyperparameter settings for TSScore.

| Parameter | Value | Description |
|---|---|---|
| ref_conf_thresh | 0.5 | Min OCR confidence for reference text |
| keep_sim_thresh | 0.5 | Min similarity to retain a detection |
| iou_link_thresh | 0.3 | IoU threshold for temporal linkage |
| target_frames | 32 | Number of sampled frames per video |
| $\alpha, \beta, \gamma$ | 1/3 each | Weights for stability components |
| clarity_$\lambda$ | 0.7 | Regularization weight for text clarity |

**CRScore.** CRScore leverages an MLLM to assess creativity and rationality. For each video, 8 frames are sampled and evaluated $N = 3$ times with different seeds. The final score is obtained by averaging these runs, mitigating stochasticity in large language model outputs.

Table 6: Hyperparameter settings for CRScore.

| Parameter | Value | Description |
|---|---|---|
| $N$ | 3 | Number of repeated MLLM evaluations per video |

## C.4 ADDITIONAL DETAILS OF QUANTITATIVE EVALUATION.

To complement the quantitative results reported in Table 3, we further provide two additional analyses. First, Figure 9 (left) presents a radar chart that jointly visualizes the six evaluation metrics. This view highlights the relative strengths of different models: closed-source systems such as Kling and Taobao Xingchen achieve higher scores across most dimensions, while open-source models lag behind, particularly in product consistency and text stability. Fine-tuned variants (LTX-InnoAds and Wan2.2-InnoAds) show clear improvements over their base models, with Wan2.2-InnoAds approaching the performance of leading closed-source systems.

Second, we conduct a user study to evaluate perceptual quality under two criteria: *prompt following* and *quality in advertising context*. As shown in Figure 9 (right), closed-source models again receive higher preference rates, with Kling 2.1 Std reaching 75% in the advertising-quality dimension. Open-source baselines remain weaker, but fine-tuned models significantly narrow the gap: Wan2.2-InnoAds achieves 63% preference in prompt following and 71% in advertising quality, surpassing its base model by a large margin. These findings confirm that while current open-source models struggle with Ad-VFX requirements, targeted fine-tuning on InnoAds-32K provides a substantial boost in both automatic and human evaluations.

## C.5 DETAILS OF INPUT PROMPTS

We design three types of input prompts to standardize the processes of video generation, evaluation, and metadata construction, as illustrated in Figure 10.

- **Video description prompts (Figure 10a).** Given an e-commerce advertising image as the first frame, the prompt instructs the model to generate a vivid and concrete video description.

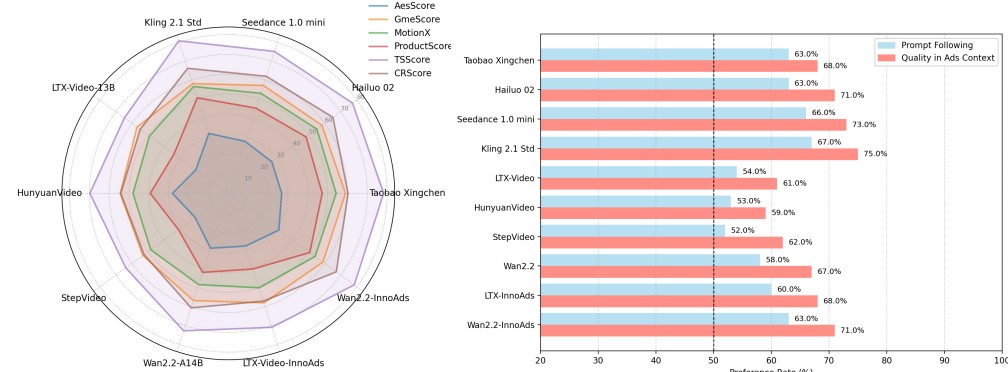

Figure 9: Quantitative comparison of baseline and fine-tuned models on InnoAds-Bench. **Left:** Radar chart illustrating six evaluation metrics (AesScore, GmeScore, MotionXScore, ProductScore, TSScore, CRScore) across all models. **Right:** Results of the user study, reporting preference rates under two criteria: prompt following and quality in advertising context.

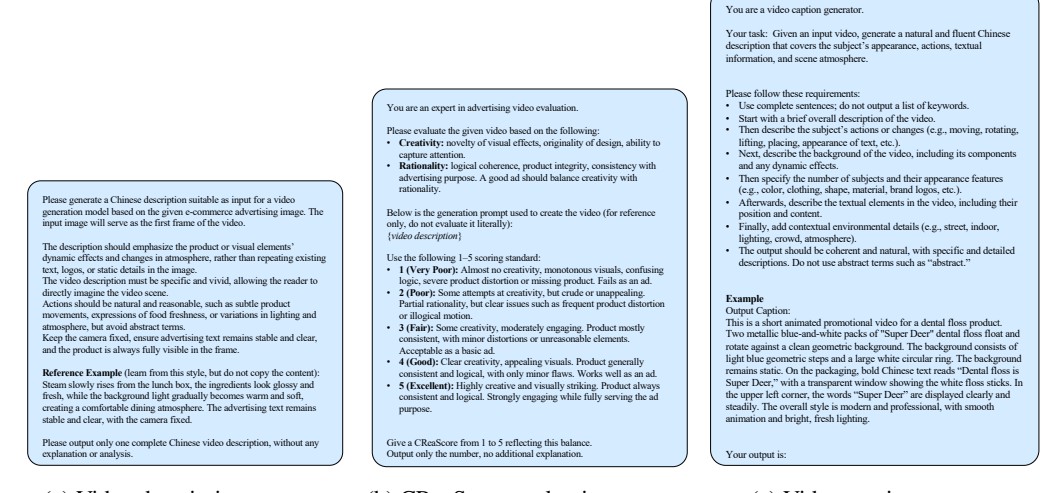

(a) Video description prompt.          (b) CReaScore evaluation prompt.          (c) Video caption prompt.

Figure 10: Prompt designs used in our framework: (a) video description prompt, (b) CReaScore evaluation prompt, (c) video caption prompt.

> It emphasizes natural product dynamics, atmosphere changes, fixed camera viewpoint, and stable advertising text, ensuring that generated videos highlight motion and scene transitions rather than static repetition.

- **CRScore evaluation prompts (Figure 10b).** To evaluate generated videos, we design a dedicated prompt for a multi-modal LLM. It guides the evaluator to assess *creativity* and *rationality* on a 1–5 scale. The model is explicitly instructed to output only the final score, enabling consistent automatic scoring.

- **Video caption prompts (Figure 10c).** For metadata enrichment, we introduce a prompt that generates natural and detailed captions for videos. These captions cover subject appearance, actions, textual content, background composition, and overall atmosphere, providing structured semantic descriptions that support downstream multi-modal research tasks.

These three prompts establish a transparent and reproducible protocol that unifies video generation, evaluation, and dataset annotation in the Ad-VFX domain. Notably, the *video description prompt* is designed primarily for video generation, focusing on potential motions, lighting, and atmosphere,

whereas the *video caption prompt* extends beyond these aspects to also include product appearance, background context, and environmental details.

# D ADDITIONAL STATEMENT

## D.1 LIMITATIONS AND FUTURE WORK

While InnoAds establishes the foundation dataset and benchmark dedicated to advertising visual effects (Ad-VFX), it still has several limitations. First, although we cover six major e-commerce domains with balanced subcategories, the dataset does not yet capture the full diversity of global advertising practices, such as regional stylistic variations or culture-specific design conventions. Second, the benchmark evaluation primarily focuses on automatically measurable dimensions; certain nuanced aspects of persuasion or emotional appeal remain difficult to capture.

Looking forward, we identify several promising directions for future research. Expanding the dataset to incorporate more diverse product categories, advertising styles, and multi-lingual contexts would further enhance its utility. On the evaluation side, integrating richer user studies and perceptual assessments could complement automatic metrics and provide deeper insights into advertising effectiveness. Finally, exploring advanced training strategies such as instruction tuning, multi-modal alignment, and reinforcement learning with human feedback may further improve model performance in Ad-VFX scenarios. We hope InnoAds will serve as a foundation for these explorations, driving progress toward robust, creative, and practically useful advertising video generation.

## D.2 DECLARATION OF LLM USAGE

We utilized Large Language Models (LLMs), including ChatGPT and related systems, during the preparation of this paper. LLMs were employed for language-related tasks such as grammar correction, spelling checks, and word choice refinement to enhance the manuscript's clarity and readability. They also supported data processing and filtering, including prompt construction for video generation, metadata captioning, and automatic evaluation components (e.g., CRScore is LLM-based). All scientific ideas, dataset construction protocols, benchmark designs, analyses, and conclusions were independently conceived, implemented, and validated by the authors.

