# OpenReview forum: "InnoAds: A Foundation Dataset and Benchmark for Advertising Visual Effects in Video Generation"
_ICLR.cc/2026/Conference — ICLR 2026 Conference Withdrawn Submission_

### Official Review · Reviewer_vLYu · 2025-10-26

**Soundness:** 2
**Presentation:** 3
**Contribution:** 2
**Rating:** 4
**Confidence:** 3

**Summary:**

The paper introduces InnoAds-32K, a large-scale dataset of over 32,000 image–text–video triples designed for advertising visual effects (Ad-VFX), and InnoAds-Bench, a benchmark evaluating six dimensions including both general and advertising-specific metrics. Through systematic evaluation, the authors show that current video generation models struggle to meet advertising needs. Together, InnoAds-32K and InnoAds-Bench establish the first standardized foundation for Ad-VFX research.

**Strengths:**

1. The paper introduces InnoAds-32K, a dataset containing over 32,000 image–text–video triples designed for advertising visual effects (Ad-VFX). Existing open-source models, such as Wan 2.2, can be fine-tuned on this dataset to achieve improved performance.

2. It also presents InnoAds-Bench, a benchmark specifically for evaluating Ad-VFX generation, featuring advertising-oriented metrics, such as MotionXScore, ProductScore, TSScore, and CRScore, to better assess the quality of generated advertising videos.

3. The paper is clearly written and easy to follow.

**Weaknesses:**

1. Although this work contributes valuable training data to the underexplored field of Ad-VFX generation, its technical novelty is limited, offering few new methodological insights.

2. The validity of the proposed metrics requires further verification. For instance, whether they align with human preferences, as discussed in [1].

3. As shown in Table 1, the dataset does not demonstrate clear advantages in scale or duration compared to existing ones.

[1] Evaluation of text-to-video generation models: A dynamics perspective[J]. Advances in Neural Information Processing Systems, 2024, 37: 109790-109816.

**Questions:**

1. Is the dataset is restricted to Chinese-language advertising scenarios?

---

### Official Review · Reviewer_9dNV · 2025-11-03

**Soundness:** 2
**Presentation:** 3
**Contribution:** 2
**Rating:** 2
**Confidence:** 4

**Summary:**

This work focuses on Advertising Visual Effects in advertising videos. Current video models struggle with this task, producing videos that exhibit issues like incoherent motion, distorted products, and unstable text, which ultimately fail to highlight the product itself. The authors attribute these shortcomings primarily to the lack of a dedicated dataset for the advertising domain and the absence of a suitable evaluation benchmark. To address this, they designed a multi-stage pipeline for dataset construction, resulting in INNOADS-32K. Furthermore, the authors adapted existing metrics to better serve the evaluation of ad videos, proposing a benchmark that includes six evaluation metrics. They evaluated multiple video models and fine-tuned two open-source models using the INNOADS-32K dataset. The experimental results demonstrate that the fine-tuned models achieve superior performance, comparable to that of closed-source models.

**Strengths:**

1. The four metrics proposed in this paper are specifically designed for Ad-VFX, with appropriate improvements to make the evaluation more reasonable. The proposed benchmark is valuable.
2. The paper evaluates a considerable number of video models and fine-tunes two open-source models on its proposed InnoAds-32K dataset, which makes the experimental section more convincing.
3. The paper is well-written and clearly details the dataset construction, benchmark creation, and experimental setup. The appendix provides sufficient implementation details.

**Weaknesses:**

1. The authors focus on the task of Advertising Visual Effects (Ad-VFX) and have constructed a specialized dataset. However, this dataset is relatively small in scale. Additionally, it is generated using video models, which means there is a gap compared to real-world advertising data, and its quality is limited by the capabilities of current advanced video models. Based on the video demos provided, I believe the degree of motion, the level of advertising creativity, and the quality of the special effects are somewhat limited.
2. The contribution of this work is limited.  I believe the Advertising Visual Effects task can be addressed by more powerful video models in the future without the need for fine-tuning on domain-specific datasets. Furthermore, as mentioned in weakness 1, the 32K dataset is generated by video models rather than being real data, and I think this dataset offers limited assistance.
3. The scope of this work is somewhat limited as it primarily focuses on advertising videos with static cameras. Real-world advertisements often include camera movements and consist of multiple shots or scenes.

**Questions:**

I believe the work would be more valuable if the authors were to focus on generating advertising videos from product images and textual information, or on creating multi-shot ad videos.

---

### Official Review · Reviewer_nMT2 · 2025-11-03

**Soundness:** 3
**Presentation:** 3
**Contribution:** 3
**Rating:** 6
**Confidence:** 3

**Summary:**

This paper addresses the significant gap between general-purpose video generation and the specific requirements of advertising visual effects (Ad-VFX). The authors make three main contributions: (1) They introduce InnoAds-32K, a new dataset of over 32,000 image-text-video triples curated for advertising scenarios. (2) They propose InnoAds-Bench, a comprehensive benchmark featuring four novel, advertising-specific metrics: $MotionXScore$, $ProductScore$, $TSScore$, and $CRScore$. (3) They provide a thorough evaluation of existing state-of-the-art (SOTA) models, demonstrating their limitations in Ad-VFX and showing that fine-tuning on InnoAds-32K yields significant improvements. However, there are some weaknesses: the synthetic dataset introduces bias due to the limitations of the models used for generation; circularity in dataset construction and benchmarking may bias evaluation results; and the failure analysis remains descriptive rather than diagnostic.

**Strengths:**

1. Clear Motivation and Problem Definition: The paper is well-motivated. It correctly identifies that SOTA video models, while impressive, fail to meet critical commercial needs for advertising, such as product consistency, motion continuity, and visual text stability. The distinction between general video generation and Ad-VFX is clear and justifies the need for new datasets and benchmarks.

2. Novel Benchmark Metrics: The strongest contribution is InnoAds-Bench. The authors introduce several novel metrics that are technically sound and directly target the identified gaps:

MotionXScore intelligently extends existing motion metrics by adding an "activation ratio" to ensure motion is persistent throughout the video, not just sporadic.

ProductScore uses a segmentation model to isolate the product before measuring similarity, making the metric more robust to background noise and text overlays common in ads.

TSScore is a comprehensive metric for an overlooked but critical problem: the stability and legibility of visual text within the video (not the prompt), measuring content stability, clarity, and temporal coverage.

3. Utility of the Dataset: Open-source models (LTX and Wan2.2) fine-tuned on InnoAds-32K outperform their base models across the new ad-specific metrics. This provides strong evidence that the dataset captures relevant features for this domain.

**Weaknesses:**

1. Dataset Construction Bias: The construction of the InnoAds-32K dataset, while a practical and innovative approach, does present a conceptual limitation. Specifically, the dataset's videos are synthetically generated using state-of-the-art models developed by Kling, Jimeng, and Wan. As a result, the dataset is inevitably influenced by the strengths and limitations inherent in these existing models, including any potential failure modes and artifacts that may arise from the generation process. This reliance on synthetic data, while useful for many purposes, does introduce an inherent bias that reflects the current capabilities and shortcomings of the generators.

2. Circularity in Dataset Construction and Benchmarking: The paper mentions in Section 3.1, under the "Video Filtering" stage, that the quantitative metrics introduced in Section 4 (i.e., the InnoAds-Bench metrics) were used to filter the generated video data. This implies that the InnoAds-32K training dataset was curated based on the final evaluation criteria. This approach creates the potential for bias, as a model trained on a dataset that has already been filtered according to the benchmark's standards is more likely to perform well on that same benchmark. The authors should address this potential circularity and discuss its implications on the fairness and validity of the evaluation results.

3. In-Depth Failure Analysis: The analysis remains largely descriptive (identifying what fails) rather than diagnostic (investigating why it fails). It would be significantly strengthened by including a discussion that analyzes the common failure modes of these SOTA models and hypothesizes their root causes.

**Questions:**

Could the authors provide a deeper, more diagnostic analysis of some causes for the failures of SOTA models in Ad-VFX scenarios?

Was any validation study conducted to correlate the MLLM's 1-5 score against human judgments of creative rationality to ensure the MLLM is aligned with human perception?

---

### Official Review · Reviewer_XeNw · 2025-11-03

**Soundness:** 2
**Presentation:** 2
**Contribution:** 2
**Rating:** 6
**Confidence:** 5

**Summary:**

This paper presents InnoAds-32K, a synthetic dataset for advertising scenario video generation. By utilizing multi-stage filtering pipeline and dedicated metrics for advertising domain, it constructs 32K high-quality 720p data with coherent motion and clear visual text. Furthermore, it collects a benchmark InnoAds-Bench, which comprises 176 cases across six major domains. Through extensive experiments, it demonstrates that fine-tuning on it yields consistent improvements on both general and advertising-specific metrics.

**Strengths:**

* By utilizing multi-stage pipeline and designing domain-specific metrics, it constructs 32K synthetic data for advertising domain, which effectively adapts foundation model on generating advertising videos.
* It constructs a diverse benchmark for evaluating the visual quality of the generated advertising videos, which fills the gap in this field.
* Extensive experiemnts, including quantitative and qualitative evaluation, demosntrates the effectiveness of the proposed dataset and benchmark.

**Weaknesses:**

* My major concern lies in the visual quality of the proposed dataset. The videos generated by foundation model often inevitably lack of detail and produce artifacts. A common solution is to collect advertising videos online and retrieval identity images by similarity metrics with other paired data. The authors need to specify the reasons of utilizing synthetic data (since it clearly lacks an advantage in both video quality and scale).
* The proposed dataset lacks of comparison with the advertising subset of OpenS2V-5M. Although comparing with several foundation models, the authors also need to filter out advertising videos from open-source subject-related datasets and perform a detailed comparison with the models fine-tuned on them.

**Questions:**

* Will the proposed dataset and benchmark be open-sourced? The authors should provide a schedule about it.
* The authors are strongly recommended to provide examples of InnoAds on supplementary material to showcase its visual quality. Also, a subset of the benchmark is encouraged to be presented. The absence of results may lead to a lower score.

---

### Official Review · Reviewer_h8xM · 2025-11-03

**Soundness:** 2
**Presentation:** 2
**Contribution:** 2
**Rating:** 4
**Confidence:** 4

**Summary:**

The paper introduces InnoAds-32K, a dataset of about 32k image–text–video triples focused on advertising visual effects (Ad-VFX), and InnoAds-Bench, a 176-case benchmark with six evaluation dimensions: two general metrics and four advertising-specific ones. The dataset is built through a multi-stage pipeline that includes SKU collection and filtering, image collection and filtering, MLLM-based prompt construction, image-to-video generation, video filtering, and metadata addition. The authors evaluate several state-of-the-art open- and closed-source models and show that (1) current models struggle with advertising-specific requirements and (2) fine-tuning on InnoAds-32K significantly improves performance on InnoAds-Bench.

**Strengths:**

The paper makes a clear and original contribution to T2V generation. Advertising videos have unique requirements such as text stability and product identity that are not well captured by existing datasets. The proposed InnoAds-32K is carefully curated and covers a wide range of products and industries.
The benchmark introduces six complementary metrics that are both technically sound and well suited to advertising applications. Metrics like ProductScore, TSScore, and MotionXScore capture fine-grained ad-specific qualities that general-purpose evaluations often ignore.
The authors evaluate multiple open- and closed-source models on InnoAds-Bench, providing a clear comparative analysis of their strengths and weaknesses.
The experiments show that fine-tuning open-source models on InnoAds-32K leads to consistent improvements across multiple metrics, which supports the usefulness of the dataset.

**Weaknesses:**

1. The dataset construction pipeline relies heavily on existing image-to-video models for generating videos, which may introduce the same biases and artifacts found in those models.  InnoAds-Bench contains 176 curated short clips of about 5 seconds each. The paper does not show how performance gains transfer to real commercial ads (e.g., longer videos, multi-shot ads, multi-product compositions). Also all generated videos are ~5s, real world ads are often much longer varying from 15 to 60 seconds.

2. Synthetic-heavy dataset and ecological validity.
The core videos are model-generated from SKU images using closed-source I2V tools (Kling, Jimeng, Wan) with professional designer curation (Figure 2), rather than real ad footage. This risks baking generator biases into both training and evaluation, and it weakens claims about real-world ad performance. Consider a parallel “real-ad” subset to validate the metrics and conclusions.
The paper gives a prompt and N, but there’s no inter-run variance or agreement analysis vs human raters.

3. Although the paper presents InnoAds as a general foundation for advertising visual effects, the dataset and experiments appear to focus exclusively on Chinese ads and Chinese-developed models. The visuals in Figures 1, 5, and 6 clearly show Chinese-language content. This is mentioned briefly in Appendix D.1, but it should be discussed more clearly in the main paper as a limitation. The authors should clarify that InnoAds primarily represents the Chinese advertising domain. Similarly, will TSScore transfer to English/Arabic scripts, stylized fonts, curved baselines, or animated text effects common in Western ads?


4. The user study section only reports overall preference scores but omits important details such as the number of annotators, their backgrounds, annotation guidelines, inter-annotator agreement, and significance testing. The prompts or instructions given to annotators are also missing.

5.   The claim that "InnoAds-32K and InnoAds-Bench, the first dataset–benchmark suite dedicated to Ad-VFX" is misleading. There is a lot of  Previous work on ad generation and evaluation benchmarks consisting of real dataset. They should be used for real world validation, such as:
- Harini, S.I., Singh, S., Singla, Y.K., Bhattacharyya, A., Baths, V., Chen, C., Shah, R.R., and Krishnamurthy, B. (2025). Long-Term Ad Memorability: Understanding & Generating Memorable Ads. In WACV 2025 (pp. 5707–5718). IEEE.
- A Video Is Worth 4096 Tokens: Verbalize Story Videos To Understand Them In Zero Shot, EMNLP

6. How many annotators participated in the user study, and what guidelines were provided? What was the inter-annotator agreement?
In the filtering process described in lines 191–195, was filtering done through an MLLM or manually by humans? What were the original image sources (for example, which e-commerce platforms)? More details on image selection and filtering would be helpful.
The TSScore calculation excludes the product region to avoid measuring logos. However, in many ads such as food or apparel, text on the product is part of the main message. Is this aspect captured by ProductScore?


6. Ads is a wide domain - what the paper does is ads of physical products, categories such as services, digital products, B2B products, etc are completely missed. Generalization beyond “product-in-frame” ads is needed. Many modern ads feature people, scenes, UGC aesthetics, and fast cuts. The dataset emphasizes fixed camera, stable text, product always visible (Appendix prompts), which could over-constrain learned behaviors and penalize valid creative ads that break these rules.

7. SKUs and ad images are collected from e-commerce platforms; licensing terms for redistribution and derivative video generation aren’t spelled out, nor are brand/identity usage constraints—important for a dataset intended as a “foundation” resource.

**Questions:**

NA

---

### Note · Authors · 2025-11-12

I have read and agree with the venue's withdrawal policy on behalf of myself and my co-authors.

---

### Note · Authors · 2025-11-12

I have read and agree with the venue's withdrawal policy on behalf of myself and my co-authors.